# The Role of Thermoviscous and Thermocapillary Effects in the Cooling and Gravity-Driven Draining of Molten Free Liquid Films

Hani Alahmadi [1,†] and Shailesh Naire [2,*,†] 

[1] Department of Mathematics, College of Science, Jouf University, Sakaka P.O. Box 2014, Saudi Arabia; hnalahmadi@ju.edu.sa
[2] School of Computing and Mathematics, Keele University, Keele ST55BG, UK
[*] Correspondence: s.naire@keele.ac.uk
[†] These authors contributed equally to this work.

**Abstract:** We theoretically considered two-dimensional flow in a vertically aligned thick molten liquid film to investigate the competition between cooling and gravity-driven draining, which is relevant in the formation of metallic foams. Molten liquid in films cools as it drains, losing its heat to the surrounding colder air and substrate. We extended our previous model to include non-isothermal effects, resulting in coupled non-linear evolution equations for the film's thickness, extensional flow speed and temperature. The coupling between the flow and cooling effect was via a constitutive relationship for temperature-dependent viscosity and surface tension. This model was parameterized by the heat transfer coefficients at the film–air free surface and film–substrate interface, the Péclet number, the viscosity–temperature coupling parameter and the slope of the linear surface tension–temperature relationship. A systematic exploration of the parameter space revealed that at low Péclet numbers, increasing the heat transfer coefficient and gradually reducing the viscosity with temperature was conducive to cooling and could slow down the draining and thinning of the film. The effect of increasing the slope of the surface tension–temperature relationship on the draining and thinning of the film was observed to be more effective at lower Péclet numbers, where surface tension gradients in the lamella region opposed the gravity-driven flow. At higher Péclet numbers, though, the surface tension gradients tended to enhance the draining flow in the lamella region, resulting in the dramatic thinning of the film in the later stages.

**Keywords:** thin film viscous flows; thermoviscous; thermocapillary

## 1. Introduction

Foams play crucial roles in a variety of applications, such as the fabrication of metallic foams [1,2] and in the food industry (e.g., bread dough) [3]. They contribute to the mechanical properties of metallic foams by enhancing their stiffness and energy absorption and are ideal for applications in the automobile industry, for example. They also contribute to the texture, aroma and visual appearance of food foams [3]. Therefore, understanding the factors that influence a foam's structure, stability and lifetime is of considerable interest.

The structures of metallic foams are broadly similar to those of aqueous foams, which are characterized by networks of thin liquid films (lamellae) intertwined with gas bubbles. The process of liquid drainage in Plateau borders and, consequently, the thinning of lamella are important in understanding bubble collapse and predicting the lifetime of a foam or its overall stability. This process has been well-studied in aqueous foams [4], where surfactants are required to stabilize foams by reducing the surface tension of the air–liquid interface. Surfactants do not affect the surface tension of metallic foams; therefore, nano- and microparticles are often added during the foaming process to increase the effective liquid viscosity and slow down the drainage, thinning and rupture times [2,5,6]. In addition,

during metallic foam formation, solidification via the cooling of liquid metal in lamella is a race against time [7] that competes against the liquid drainage. This competition then determines the overall stability and pore structure of the metallic foam. The cooling and subsequent freezing of metallic foams have received very little attention, even though they are crucial in the manufacture of these foams.

Non-isothermal effects are important when there is a strong coupling between a flow and temperature field due to the strong dependence of liquid properties on temperature. The viscosity of most materials decreases with temperature. Some materials, such as glass, metallic melts and polymer melts, can exhibit dramatic changes in their viscosity due to variations in temperature, e.g., the cooling and solidification of silicate (or glass-like) lava flows [8]. For glass and polymers, surface tension can also vary with temperature (surface tension in most liquids decreases with an increase in temperature), although perhaps not as dramatically as viscosity.

In the context of metallic foams, the heat transfer between hot liquid within the lamella and Plateau borders and the cooler surrounding gas bubbles via the free surface could result in the lamella cooling down considerably and rapidly in some situations. The resulting thermoviscous (viscosity variations with temperature) and thermocapillary effects (surface tension variations with temperature) could have significant influence on film drainage and thinning and overall foam stability.

Indeed, Cox et al. [7] were the first to theoretically investigate the competition between liquid drainage and freezing in the formation of metallic foams. They combined the so-called foam drainage equation [4] with the heat conduction equation to derive a bubble coalescence criterion, which allows for the rupture of thin films. Their one-dimensional model is restricted to cooling that takes place at the top and bottom surfaces and does not account for heat loss from the air–liquid interface. Moreover, they only investigated viscosity variations with temperature and not surface tension variations. More recently, Shah et al. [9] investigated the influence of thermal fluctuations on the drainage, thinning and rupture of liquid films. They showed that thickness variations due to thermal fluctuations at the free surface (originating from the random thermal motion of molecules) can compete with curvature-induced drainage at Plateau borders. In particular, when drainage is weak, the film ruptures at a random location due to the spontaneous growth of fluctuations originating from thermal fluctuations. This is in contrast to the scenario where drainage is strong, which results in the film rupturing at a local depression (a so-called dimple) between the lamella and Plateau border. It is worth mentioning that the roles of thermoviscous and thermocapillary effects have also been investigated in the related context of extensional flows associated with the drawing of viscous threads or sheets, focusing on the stretching and pinching of threads [10,11] or sheet rupture [12,13]. The goal of this paper was to fully investigate the coupling between gravity-driven extensional flows and cooling, without the limitations imposed by Cox et al. [7]; while we did not consider phase transition due to freezing, we accounted for cooling from both the air–liquid interface and the top and bottom surfaces. Moreover, we considered the influence of both thermoviscous and thermocapillary effects on the drainage and cooling of molten liquid films.

The outline of this paper is as follows. We formulate the two-dimensional mathematical problem in Section 2, which provides the governing equations and boundary conditions for the flow and temperature field. The lubrication approximation using the fact that the film's aspect ratio is small allowed for the simplification of the governing equations and boundary conditions in a system of three coupled PDEs for the evolution of one-dimensional free-surface shapes and extensional flow speeds, as well as two-dimensional temperature fields. In Section 4, we perform numerical simulations of the evolution equations to determine the free surface shapes, extensional flow speeds and temperature fields for a variety of parameter values related to the Péclet number, heat transfer coefficients, an exponential viscosity–temperature model and a linear surface tension–temperature model. In Section 5, we discuss the main results.

## 2. Methods

Following previous work [14,15], we consider the two-dimensional flow due to the draining of a liquid in a vertically-aligned film. The film has two free surfaces and is suspended between two horizontal solid frames as shown in Figure 1. The liquid in the film is hot, at an initial temperature $T_i^*$, compared to its cooler surroundings which are at ambient temperature $T_a^*$. The configuration shown in Figure 1 mimics the thinning of a lamella draining into a Plateau border and is a simple idealization of a liquid foam film. Other configurations that have been investigated include a film suspended over a liquid bath [16–20]. It is much simpler to prescribe boundary conditions at the upper and lower ends in the configuration considered here. In addition, we assume that the film is drawn out sufficiently quickly for a stable initial film profile to exist. The speed at which the film is drawn will influence its stability and whether a film of a specified height and thickness can be achieved [16].

The initial liquid film is sufficiently thick for gravity to play a significant role in its drainage. The liquid loses its heat via the cooler free surface at $z^* = h^*(x^*, t^*)$ and the top and bottom supports at $x^* = 0, L^*$. The flow evolves due to the effects of gravity, viscous forces and surface tension causing the liquid in the film to drain downwards, resulting in the thinning of the film. The liquid is assumed to be an incompressible and Newtonian liquid with constant properties, except the liquid viscosity and surface tension, which are dependent on the temperature. We do not consider phase transition associated with solidification due to freezing near the surface or supports. The ambient temperature $T_a^*$ is assumed to be much higher than the melting point to prevent the film from freezing.

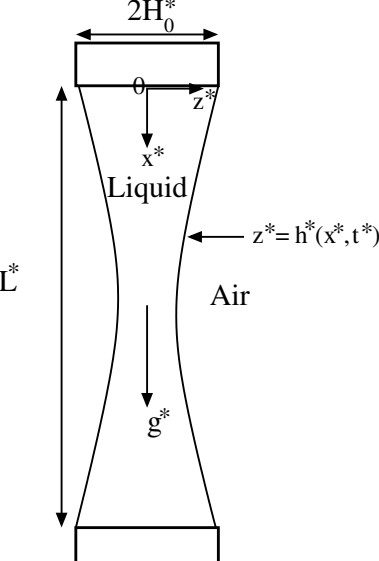

**Figure 1.** Schematic of a vertically-aligned two-dimensional free liquid film draining under gravity between two rigid frames (adapted from Alahmadi et al. [14]). The liquid within the film is hot compared to its cooler surroundings.

Figure 1 shows a schematic of the geometry. We consider a two-dimensional Cartesian coordinate system $(x^*, z^*)$ with the $x^*$-axis along the film's length and the $z^*$-axis along the film's thickness. The horizontal frames are separated by a distance $L^*$ and have width $2H_0^*$. We assume symmetry about the film's centre line at $z^* = 0$. The two free surfaces of the film are represented by $z^* = \pm h^*(x, t)$. Assuming left–right symmetry, we only consider half of the film between $z^* = 0$ and $z^* = h^*(x, t)$. The superscript $^*$ refers to dimensional quantities.

### 2.1. Governing Equations

The flow is described by the Navier–Stokes equations. The density $\rho^*$ is assumed constant, so the continuity equation reduces to

$$u_{x^*}^* + w_{z^*}^* = 0. \tag{1}$$

In the above, $\mathbf{v}^* = (u^*, w^*)$ are the flow speeds in the $x^*$ and $z^*$ directions, respectively, and the subscript denotes differentiation with respect to the subscript variable. The momentum equations can be written as:

$$\rho^*(u_{t^*}^* + u^* u_{x^*}^* + w^* u_{z^*}^*) = -p_{x^*}^* + \tau_{x^*}^{*xx} + \tau_{z^*}^{*xz} + \rho^* g^*, \tag{2a}$$

$$\rho^*(w_{t^*}^* + u^* w_{x^*}^* + w^* w_{z^*}^*) = -p_{z^*}^* + \tau_{x^*}^{*xz} + \tau_{z^*}^{*zz}, \tag{2b}$$

where $p^*$ is the liquid pressure, $\tau^{*xx}$ and $\tau^{*zz}$ are the extensional viscous stresses in the $x^*$ and $z^*$ directions, respectively, $\tau^{*xz}$ is the viscous shear stress and $g^*$ is the acceleration due to the gravity.

The constitutive relation between the viscous stress $\boldsymbol{\tau}^*$ and the shear rate $\dot{\boldsymbol{\gamma}}^*$ for a Newtonian liquid with temperature-dependent viscosity is written as:

$$\boldsymbol{\tau}^* = \mu^*(T^*)\dot{\boldsymbol{\gamma}}^*, \tag{3}$$

where $\mu^*(T^*)$ is the temperature-dependent liquid viscosity, $T^*$ is the temperature, and

$$\boldsymbol{\tau}^* = \begin{pmatrix} \tau^{*xx} & \tau^{*xz} \\ \tau^{*xz} & \tau^{*zz} \end{pmatrix}, \quad \dot{\boldsymbol{\gamma}}^* = \begin{pmatrix} 2u_{x^*}^* & u_{z^*}^* + w_{x^*}^* \\ u_{z^*}^* + w_{x^*}^* & 2w_{z^*}^* \end{pmatrix}, \tag{4}$$

The two-dimensional governing equation for the temperature, $T^*$ in Cartesian coordinates, $(x^*, z^*)$ is given by

$$\rho^* c_p^*(T_{t^*}^* + u^* T_{x^*}^* + w^* T_{z^*}^*) = \kappa^*[T_{x^* x^*}^* + T_{z^* z^*}^*], \tag{5}$$

in a material with density $\rho^*$, specific heat $c_p^*$, thermal conductivity $\kappa^*$ and thermal diffusivity $\kappa_d^* = \kappa^*/(\rho^* c_p^*)$. These are assumed to be constant and independent of temperature. We neglect the contribution from viscous dissipation.

### 2.2. Boundary Conditions

Symmetry along the center line $z^* = 0$ is imposed through the boundary conditions:

$$w^* = u_{z^*}^* = \tau^{*xz} = T_{z^*}^* = 0, \text{ at } z^* = 0. \tag{6}$$

At the free surface $z^* = h^*(x^*, t^*)$, we have the stress boundary conditions normal and tangential to the free surface. The normal stress boundary condition balances the jump in the total normal stress (between the outside air and the liquid) with the product of the surface tension times and the curvature of the free surface,

$$-p^* + \frac{1}{1 + h_{x^*}^{*2}}\left[h_{x^*}^{*2}\tau^{*xx} - 2h_{x^*}^*\tau^{*xz} + \tau^{*zz}\right] = \frac{\sigma^*(T^*)h_{x^* x^*}^*}{\left(1 + h_{x^*}^{*2}\right)^{\frac{3}{2}}}, \tag{7}$$

where $\sigma^*(T^*)$ is the temperature-dependent surface tension and $h_{x^* x^*}^* / \left(1 + h_{x^*}^{\star 2}\right)^{\frac{3}{2}}$ is the surface curvature. Without loss of generality, we take the atmospheric pressure to be zero, therefore, the liquid pressure $p^*$ is relative to the atmospheric pressure. The tangential stress at the free surface for the non-isothermal case is driven by gradients in surface tension due to variations in temperature (the so-called Marangoni stress). The tangential stress boundary condition can be written as:

$$(1 - h_{x^*}^{*2})\tau^{*xz} + h_{x^*}^* \left(\tau^{*zz} - \tau^{*xx}\right) = [\sigma_{x^*}^*(T^*) + h_{x^*}^* \sigma_{z^*}^*(T^*)]\sqrt{1 + h_{x^*}^{*2}}. \tag{8}$$

At the free surface $z^* = h^*(x^*, t^*)$, we also impose a heat flux boundary condition based on Newton's law of cooling which assumes that the heat flux is proportional to the temperature difference across this boundary. This is written as:

$$-\kappa^* \mathbf{n}^* \cdot \nabla T^* = a_m^*(T^* - T_a^*), \tag{9}$$

where $a_m^*$ is a heat transfer coefficient (assumed constant) and $T_a^*$ is the ambient temperature (assumed constant), and $\mathbf{n}^* = \dfrac{1}{\sqrt{1 + h_{x^*}^{*2}}}(-h_{x^*}^*, 1)$ is the outward-pointing normal vector to the free surface. We can write Equation (9) as:

$$\kappa^* \left(1 + h_{x^*}^{*2}\right)^{-\frac{1}{2}} (T_{z^*}^* - h_{x^*}^* T_{x^*}^*) = -a_m^*(T^* - T_a^*). \tag{10}$$

Finally, the kinematic boundary condition at the free surface is given by

$$h_{t^*}^* = w^* - u^* h_{x^*}^*, \text{ at } z^* = h^*(x^*, t^*). \tag{11}$$

At the top and bottom boundary $x^* = 0, L^*$, respectively, the film is pinned to the end of the frame and we impose no slip,

$$h^* = H_0^* \text{ and } \mathbf{v}^* = 0, \text{ at } x^* = 0, L^*. \tag{12}$$

Here we also impose the following heat flux boundary condition:

$$-\kappa^* \mathbf{n}^* \cdot \nabla T^* = b_s^*(T^* - T_s^*), \tag{13}$$

$$\begin{cases} \kappa^* T_{x^*}^* = b_s^*(T^* - T_s^*), & \text{at } x^* = 0, \\ -\kappa^* T_{x^*}^* = b_s^*(T^* - T_s^*), & \text{at } x^* = L^*, \end{cases} \tag{14}$$

where $b_s^*$ is a heat transfer coefficient at the wire frames (assumed constant) and $T_s^*$ is the temperature there (assumed constant). In the above, we have used the fact that $\mathbf{n}^* = (-1, 0)$ at $x^* = 0$ and $\mathbf{n}^* = (1, 0)$ at $x^* = L^*$.

Using Equation (1) and applying Leibniz's rule, one can re-write the kinematic boundary condition, Equation (11), as

$$h_{t^*}^* + Q_{x^*}^* = 0, \quad Q^* = \int_0^{h^*} u^*(x^*, z^*, t^*) \, dz^*, \tag{15}$$

where $Q^*(x^*, t^*)$ is the liquid flux at any location $x^*$ along the length of the film. Equation (15) represents the evolution of the film thickness, $h^*(x^*, t^*)$.

The flow is coupled to the temperature field via a constitutive relationship between the viscosity and temperature $\mu^*(T^*)$ and the surface tension and temperature $\sigma^*(T^*)$. We assume an exponential decay in viscosity with temperature [21] and a linear dependence of surface tension on temperature [12] to describe this relationship, given by:

$$\mu^* = \mu_{min}^* + (\mu_0^* - \mu_{min}^*)e^{-\alpha^*(T^* - T_a^*)}, \tag{16a}$$
$$\sigma^* = \sigma_0^* - M^*(T^* - T_a^*), \tag{16b}$$

where $\alpha^*$ is a temperature–viscosity coupling constant, $\mu_0^*$ is a reference viscosity (at temperature $T_a^*$), $\mu_{min}^*$ is a minimum viscosity limit, $M^* = \dfrac{d\sigma^*}{dT^*}\big|_{(\sigma_0^*, T_a^*)}$ is the rate at which surface tension depends linearly on temperature and $\sigma_0^*$ is a reference surface tension (at temperature $T_a^*$).

Table 1 shows the physical quantities appearing in the model including their estimated values. The melt properties are based on aluminium foam melts (Tripathi et al. [22] and references therein).

**Table 1.** Physical quantities in the model. The liquid melt properties and temperatures are based on Aluminium melts (Tripathi et al. [22] and references therein).

| Physical Quantity | Estimated Value |
|---|---|
| initial temperature, $T_i^*$ | 700–800 °C |
| ambient temperature, $T_a^*$ | >660 °C (melting point) |
| temperature drop, $T_i^* - T_a^*$ | 40–140 °C (based on melting point 660 °C) |
| temperature at wire frames, $T_s^*$ | $T_a^*$ (assumed) |
| density at $T_a^*$, $\rho^*$ | $2.7 \times 10^3$ kg/m$^3$ |
| viscosity at $T_a^*$, $\mu_0^*$ | 1 Pa s (generally 1–1.4 mPa s but assumed to be enhanced by addition of particles [2,5,6] |
| minimum viscosity limit, $\mu_{min}^*$ | $\mu_0^*/10$ Pa s (assumed) |
| surface tension at $T_a^*$, $\sigma_0^*$ | 850–1100 mN/m |
| speciic heat capacity, $c_p^*$ | 0.9 kJ/kg K |
| thermal conductivity, $\kappa^*$ | 237 W/m K |
| thermal diffusivity, $\kappa_d^* = \kappa^*/(\rho^* c_p^*)$ | $9.7 \times 10^{-5}$ m$^2$/s |
| free surface heat transfer coefficient, $a_m^*$ | 1–10$^3$ W/m$^2$ K (assumed) |
| wire frame heat transfer coefficient, $b_s^*$ | $a_m^*$ (assumed) |
| temperature–viscosity coupling constant, $\alpha^*$, | 0.01–0.5 °C$^{-1}$ (based on viscosity drop from $\mu_0^*$ to $\mu_{min}^*$ in temperature range $T_i^*$ to $T_a^*$) |
| slope of surface tension–temperature relationship, $M^*$, | 10$^{-6}$–10$^{-5}$ N/m°C (based on 0.01% drop in surface tension in temperature range $T_i^*$ to $T_a^*$) |
| characteristic film length, $L^*$ | 10$^{-2}$ m |
| characteristic film thickness, $H_0^*$ | 50 μm |
| characteristic flow speed, $U^* = \dfrac{\rho^* g^* L^{*2}}{\mu_0^*}$ | 2.7 m/s |
| characteristic pressure, $p^* = \rho^* g^* L^*$ | 270 N/m$^2$ |
| characteristic time, $t^* = \dfrac{L^*}{U^*}$ | 4 ms |

### 2.3. Nondimensionalization of the Governing Equations and Boundary Conditions

We focus on the scenario where the flow is primarily extensional (or plug flow) and there is a balance between extensional viscous stresses and gravity. Following Alahmadi and Naire [14], the appropriate nondimensionalization is:

$$x^* = L^* x, \quad (z^*, h^*) = H_0^*(z, h), \quad (u^*, w^*) = \frac{\rho^* g^* L^{*2}}{\mu_0^*}(u, \epsilon w),$$

$$(p^*, \tau^{*xx}, \tau^{*zz}, \tau^{*xz}) = \rho^* g^* L^*(p, \tau^{xx}, \tau^{zz}, \frac{1}{\epsilon}\tau^{xz}),$$

$$(\gamma^{*xx}, \gamma^{*zz}, \gamma^{*xz}) = \mu_0^* \rho^* g^* L^*(\gamma^{xx}, \gamma^{zz}, \frac{1}{\epsilon}\gamma^{xz}),$$

$$t^* = \frac{\mu_0^*}{\rho^* g^* L^*} t, \quad Q^* = \frac{\rho^* g^* L^{*2}}{\mu_0^*} H_0^* Q,$$

$$T^* = T_a^* + (T_i^* - T_a^*)\theta, \quad (0 \leq \theta \leq 1). \tag{17}$$

where $\theta = 0$ implies $T^* = T_a^*$ and $\theta = 1$ implies $T^* = T_i^*$. The ratio of the two length scales is denoted by $\epsilon = \dfrac{H_0^*}{L^*}$, which is typically much less than one. We are interested in deriving the thin film equations in the asymptotic limit $\epsilon \to 0$.

Substituting Equation (17) into the governing equations and boundary conditions gives the following nondimensionalized system:

$$u_x + w_z = 0, \tag{18a}$$

$$\epsilon^2 Re(u_t + uu_x + wu_z) = -\epsilon^2 p_x + \epsilon^2 \tau_x^{xx} + \tau_z^{xz} + \epsilon^2, \tag{18b}$$

$$\epsilon^2 Re(w_t + uw_x + ww_z) = -p_z + \tau_x^{xz} + \tau_z^{zz}, \tag{18c}$$

$$Pe_r[\theta_t + u\theta_x + w\theta_z] = \epsilon^2\theta_{xx} + \theta_{zz}, \tag{18d}$$

$$\begin{pmatrix} \tau^{xx} & \tau^{xz} \\ \tau^{xz} & \tau^{zz} \end{pmatrix} = \mu(\theta) \begin{pmatrix} 2u_x & u_z + \epsilon^2 w_x \\ u_z + \epsilon^2 w_x & 2w_z \end{pmatrix}, \tag{18e}$$

$$w = u_z = \tau^{xz} = \theta_z = 0, \text{ at } z = 0, \tag{18f}$$

$$\frac{\epsilon}{\hat{Ca}} \frac{\sigma(\theta)h_{xx}}{(1 + \epsilon^2 h_x^2)^{\frac{3}{2}}} = -p +$$

$$\frac{1}{1 + \epsilon^2 h_x^2}\left[\epsilon^2 h_x^2 \tau^{xx} - 2h_x \tau^{xz} + \tau^{zz}\right], \text{ at } z = h(x,t), \tag{18g}$$

$$(1 - \epsilon^2 h_x^2)\tau^{xz} + \epsilon^2 h_x(\tau^{zz} - \tau^{xx}) =$$

$$\frac{\epsilon}{\hat{Ca}}\left[\sigma_x(\theta) + h_x \sigma_z(\theta)\right]\sqrt{1 + \epsilon^2 h_x^2}, \text{ at } z = h(x,t), \tag{18h}$$

$$\theta_z = \epsilon^2 h_x \theta_x - a\epsilon^2\theta\sqrt{1 + \epsilon^2 h_x^2}, \ z = h(x,t), \tag{18i}$$

$$h_t + Q_x = 0, \quad Q = \int_0^h u(x,z,t)\,dz, \tag{18j}$$

$$h = 1, u = w = 0, \text{ at } x = 0,1, \tag{18k}$$

$$\theta_x = \epsilon^2 b(\theta - \theta_s), \text{ at } x = 0, \tag{18l}$$

$$\theta_x = -\epsilon^2 b(\theta - \theta_s), \text{ at } x = 1, \tag{18m}$$

$$\mu(\theta) = \mu_{min} + (1 - \mu_{min})e^{-\alpha\theta}, \quad \sigma(\theta) = 1 - \epsilon^2 M\theta. \tag{18n}$$

In the above, the dimensionless number $Re = \dfrac{\rho^* U^{*2}/L^*}{\mu_0^* U^*/L^{*2}}$ is the Reynolds number (which compares inertial and extensional viscous forces with $U^* = \dfrac{\rho^* g^* L^{*2}}{\mu_0^*}$), $\hat{Ca} = \dfrac{\mu_0^* U^*}{\sigma_0^*}$ is the capillary number (which compares extensional viscous and surface tension forces), the reduced Péclet number $Pe_r = \epsilon^2 Pe$, $Pe = (\rho^* c_p^* U^* L^*)/\kappa^* = U^* L^*/\kappa_d^*$ is the Péclet number (which compares convective to diffusive heat transport), $\alpha = \alpha^*(T_i^* - T_a^*)$ is a temperature–viscosity coupling constant, $\mu_{min} = \mu_{min}^*/\mu_0^*$, $M = [M^*(T_i^* - T_a^*)/\sigma_0^*]/\epsilon^2$ is the rate of decrease in surface tension with temperature, $a = a_m^* H_0^*/(\epsilon^2\kappa^*)$ and $b = b_s^* H_0^*/(\epsilon^2\kappa^*)$ are the heat transfer coefficients at the free surface and substrate, respectively, and $\theta_s = (T_s^* - T_a^*)/(T_i^* - T_a^*)$. We will see later on that surface tension effects will be important over smaller lengthscales; in anticipation of this, we define a rescaled capillary number $Ca = \dfrac{\mu_0^* U^*}{\epsilon\sigma_0^*} = \hat{Ca}/\epsilon$, $\hat{Ca} = O(1)$, and retain the surface tension term at leading order. We assume $(Pe_r, M, a, b) = O(1)$.

Table 2 shows the dimensionless parameters appearing in the model and their estimated values.

**Table 2.** Dimensionless parameters in the model and their estimated values.

| Dimensionless Parameters | Values |
|---|---|
| Aspect ratio, $\epsilon = H_0^*/L^*$ | $5 \times 10^{-3}$ |
| Reynolds number, $Re = \dfrac{\rho^* U^* L^*}{\mu^*}$ | 72 |
| Capillary number, $\hat{C}a = \dfrac{\mu_0^* U^*}{\sigma_0^*}$ | 0.27–2.7 |
| Rescaled Capillary number, $Ca = \hat{C}a/\epsilon$ | 540–5400 |
| Péclet number, $Pe = U^* L^*/\kappa_d^*$ | $10^2$ |
| Reduced Péclet number, $Pe_r = \epsilon^2 Pe$ | $2.5 \times 10^{-3}$ |
| Temperature–viscosity coupling, $\alpha = \alpha^*(T_i^* - T_a^*)$ | 0.4–70 |
| Minimum viscosity, $\mu_{min} = \mu_{min}^*/\mu_0^*$ | $10^{-1}$ |
| Rescaled surface tension–temperature slope, $M = [M^*(T_i^* - T_a^*)/\sigma_0^*]/\epsilon^2$ | 0.04–0.1 |
| Rescaled heat transfer coefficients, $(a, b) = (a_m^*, b_s^*) H_0^*/(\epsilon^2 \kappa^*)$ | $10^{-2}$–10 |
| Wire frame temperature, $\theta_s = (T_s^* - T_a^*)/(T_i^* - T_a^*)$ | 0 |

*2.4. The Small Aspect Ratio, $\epsilon = \dfrac{H_0^*}{L^*} \ll 1$, Approximation*

We exploit the fact that $\epsilon = \dfrac{H_0^*}{L^*} \ll 1$ and expand each of the unknowns variables $(u, w, p, \tau^{xx}, \tau^{zz}, \tau^{xz})$ as a power series in $\epsilon^2$ of the form:

$$(u, w, p, \tau^{xx}, \tau^{zz}, \tau^{xz}, \theta) = (u, w, p, \tau^{xx}, \tau^{zz}, \tau^{xz}, \theta)_0(x, z, t) + \epsilon^2 (u, w, p, \tau^{xx}, \tau^{zz}, \tau^{xz}, \theta)_1(x, z, t) + O(\epsilon^4). \tag{19}$$

Substituting this in Equation (18), we can sequentially solve for the $O(1)$ and $O(\epsilon^2)$ quantities, using which the PDEs and boundary conditions describing the evolution of the film's free surface $h(x, t)$ and the extensional flow speed $u_0(x, t)$ can be derived as the leading order. The details of the derivation are provided in Appendix A. The system of PDEs and boundary conditions are given by the following (for simplicity, we drop the subscript 0):

$$h_t + Q_x = 0, \quad Q = uh, \tag{20a}$$

$$Re\, h(u_t + uu_x) - 4\left[\mu(\theta)h_x u_x + \int_0^h (\mu(\theta)u_x)_x \, dz\right] = h\left[\frac{1}{Ca}h_{xxx} + 1\right] - \frac{M}{Ca}[\theta_x + h_x\theta_z|_{z=h}], \tag{20b}$$

$$\mu(\theta) = \mu_{min} + (1 - \mu_{min})e^{-\alpha\theta}, \tag{20c}$$

$$Pe_r[\theta_t + u\theta_x + w\theta_z] = \epsilon^2\theta_{xx} + \theta_{zz}, \quad w(x, z, t) = -u_x z, \tag{20d}$$

$$\theta_z = -a\epsilon^2\theta, \text{ at } z = h(x, t), \; \theta_z = 0, \text{ at } z = 0, \tag{20e}$$

$$\theta_x = \epsilon^2 b(\theta - \theta_s), \text{ at } x = 0, \; \theta_x = -\epsilon^2 b(\theta - \theta_s), \text{ at } x = 1 \tag{20f}$$

$$h(0, t) = h(1, t) = 1, h_{xxx}(0, t) = h_{xxx}(1, t) = -Ca, \tag{20g}$$

$$u(0, t) = u(1, t) = 0. \tag{20h}$$

The boundary conditions in Equations (20g) and (20h) correspond physically to the film being pinned at the top and bottom (first two boundary conditions in Equation (20g)), and no flux out of the rigid wire supports, so $Q = 0$ (represented by the last two boundary conditions in Equation (20g) and boundary conditions in Equation (20h)). As a consequence of this, both $u$ and $u_x$ are forced to be zero near the ends and the film evolves to quasi-static shapes there. We also retain the $O(\epsilon^2)$ term in Equation (20d) in order to satisfy the boundary conditions for $\theta$ at $x = 0, 1$ (boundary conditions in Equation (20f)).

## 3. Numerical Methods

Equations (20a) and (20b) for $h(x,t)$ and $u(x,t)$, respectively, are solved for $x \in [0,1]$ with boundary conditions given by Equations (20g) and (20h). The two-dimensional evolution equation, Equation (20d), for the temperature $\theta(x,z,t)$ is solved for $(x,z) \in [0,1] \times [0, h(x,t)]$ with boundary conditions given by Equations (20e) and (20f). For computational convenience, it is useful to map the temperature field $\theta(x,z,t)$ onto a rectangular domain using the change of variables $\bar{z} = z/h$. The transformed evolution equation for the temperature $\theta(x,\bar{z},t)$ is solved for $(x,\bar{z}) \in [0,1] \times [0,1]$. The transformed evolution equations for $h$, $u$ and $\theta$ are given by Equations (A22) and (A23) shown in Appendix B. In what follows, we drop the bar in $z$ with the implicit understanding that $z \in [0,1]$.

The equations are solved numerically using the method of lines on a uniform and fixed computational mesh in the spatial directions $(x,z)$ [23]. The spatial derivatives are discretised using second-order centered finite difference schemes including a first-order upwind scheme for convection terms in the temperature equation (the terms multiplying $\theta_x$ and $\theta_z$ on the left-hand-side of Equation (A22a)). The time derivatives appearing in the equations are kept continuous. We use the trapezoidal rule to approximate the integral in the expression for $u(x,t)$ in (A23b). The resulting systems of differential-algebraic equations for the unknowns in $h$, $u$ and $\theta$ at each grid point are solved in MATLAB (Release 2013a, The MathWorks Inc., Natick, MA, USA) using the stiff ODE solver *ode15i*. The corresponding computational mesh sizes were $\Delta x, z = 10^{-3}$–$10^{-2}$ resulting in a system of $O(10^4$–$10^6)$ differential-algebraic equations (DAEs) required to be solved at each time step. For $Pe_r \gg 1$, the problem can have very narrow thermal boundary layers near $z = h(x,t)$ of width $O(Pe_r^{-1/2})$ and $x = 0, 1$ of width $O(\epsilon Pe_r^{-1/2})$. The smallest value of $\Delta z = 10^{-3}$ is sufficient to resolve these boundary layers for $Pe_r \leq 10^3$. For $Pe_r > 10^3$, much smaller values of $\Delta x, z$ are required which increase the number of DAEs at each time step, hence the computational effort. These results are not shown here as they are not different from the $Pe_r = 10^3$ results. The time step was controlled within the solver to maintain the stability of the numerical solutions. The accuracy and convergence of the numerical scheme are formally checked by systematically reducing the mesh sizes $\Delta(x,z)$ for sample cases corresponding to a low, intermediate and high reduced Péclet number $Pe_r$. Based on this, we can confirm that for the mesh sizes stated above, the numerical solutions presented below are an accurate reflection of the draining process.

## 4. Results

We seek numerical solutions to the evolution of film thickness $h(x,t)$, extensional flow speed $u(x,t)$ and temperature $\theta(x,z,t)$ by varying the key parameters: the reduced Péclet number $Pe_r$ (or Péclet number $Pe$), rate of linear decrease in surface tension with temperature $M$, the heat transfer coefficients $a, b$ at the free surface and substrate, respectively, and the temperature–viscosity coupling constant $\alpha$. Table 2 provides a range of values for the dimensionless parameters. We do not restrict the choice of the values of these parameters to be based on Table 2, but allow for a full range of realistic values to be explored in $(Pe_r, M, a, b, \alpha)$ space. We consider variations in the above parameters for $Ca = 10^3$ (representative of $Ca \gg 1$) and $Re = 0$. $Re \ll 1$ has no significant influence on the evolution of the film and the extensional speed, hence we choose $Re = 0$. Additionally, we choose the heat transfer coefficient at the top and bottom ends $b = 0$ focusing on $a$ and the heat transfer coefficient at the free surface only. The initial condition is $h(x,0) = \theta(x,z,0) = 1$ and the corresponding initial condition for the extensional flow speed is $u(x,0) = x(1-x)/8$, obtained by solving Equation (20b) for $(h, \theta) = 1$ and $Re = 0$.

We first investigate the influence of viscosity varying with temperature, and take the surface tension to be constant ($M = 0$). The solid curves in Figure 2a,e show that the evolution of $h(x,t)$ ($h(x,t)$ is plotted on a logarithmic scale) for $t = 0$–160 (in steps of 20) with $\mu = 1$ (or $\theta = 0$ everywhere corresponding to a film with liquid at the ambient temperature $T_a^*$) and $\mu_{min} = 5 \times 10^{-2}$ (or $\theta = 1$ corresponding to a film with liquid at a hotter temperature $T_i^*$ everywhere), respectively. Both these cases are isothermal with

differing liquid viscosities. The solid curves in Figure 2b,f show the extensional speed $u(x,t)$ corresponding to $\mu = 1, \mu_{min}$, respectively. The remaining curves in Figure 2a,c,e show the evolution of $h(x,t)$ ($h(x,t)$ is plotted on a logarithmic scale) for $t = 0$–160 (in steps of 20) for $Pe_r = 10^{-1}, 10, 10^2$ and $10^3$, respectively, with fixed $\alpha = 2$, $a = 0.02$, $Ca = 10^3$ and $Re = 0$. Figure 2b,d,f show the corresponding evolution of $u(x,t)$, respectively.

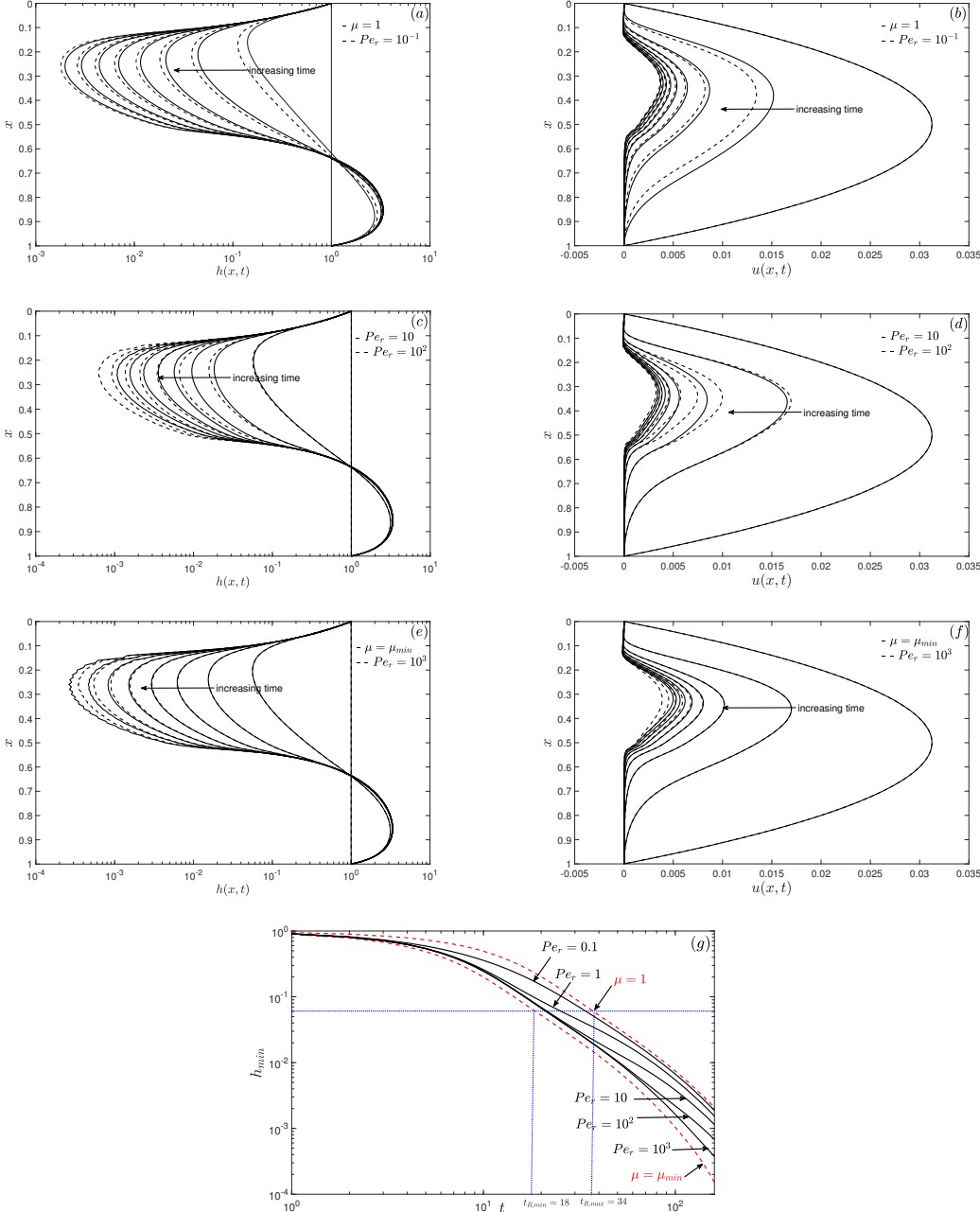

**Figure 2.** The evolution of the film thickness $h(x,t)$ (on a logarithmic scale) for $t$ varying between $t = 0$–160 (in steps of 20) corresponding to (**a**) $\mu = 1$ (solid curves; isothermal case with $\theta = 0$ everywhere) and $Pe_r = 10^{-1}$ (dashed curves), (**c**) $Pe_r = 10$ (solid curves) and $Pe_r = 10^2$ (dashed curves) and (**e**) $Pe_r = 10^3$ (dashed curves) and $\mu = \mu_{min} = 5 \times 10^{-2}$ (solid curves; isothermal case with $\theta = 1$). The corresponding extensional flow speed $u(x,t)$ is shown in (**b**,**d**,**f**). The evolution (**g**) of the global minimum $h_{min}$ as a function of time $t$ for varying $Pe_r$. The parameter values are: $\alpha = 2$, $a = 0.02$, $Ca = 10^3$ and $Re = 0$.

At early times, the fluid in the film drains downwards leading to thinning of the film in the upper region and a thickening in the lower region, and the film shape is concave-out (Figure 2a,c,e; see also the outline profile for $h$ shown in the leftmost panel in Figure 3a,d,g,j). At late times, the fluid has drained significantly towards the lower end of the domain forming a quasi-static pendant drop there, leaving a very thin and almost flat film (lamella) in the middle region, and a quasi-static capillary meniscus at the upper end (Figure 2a,c,e; see also the outline profile for $h$ shown in the rightmost panel in Figure 3c,f,i,l). This late-time behaviour can be clearly observed using a logarithmic scale for $h(x,t)$ shown in Figure 2a,c,e. This shows the middle lamella region connecting onto quasi-static curves at the top and bottom represented by the capillary meniscus and the pendant drop, respectively. The maximum flow speeds are in the middle lamella section of the film (Figure 2b,d,f) which causes the film thickness to decrease severely there. The flow speed is zero near the top in the capillary meniscus region and at the bottom in the pendant drop region.

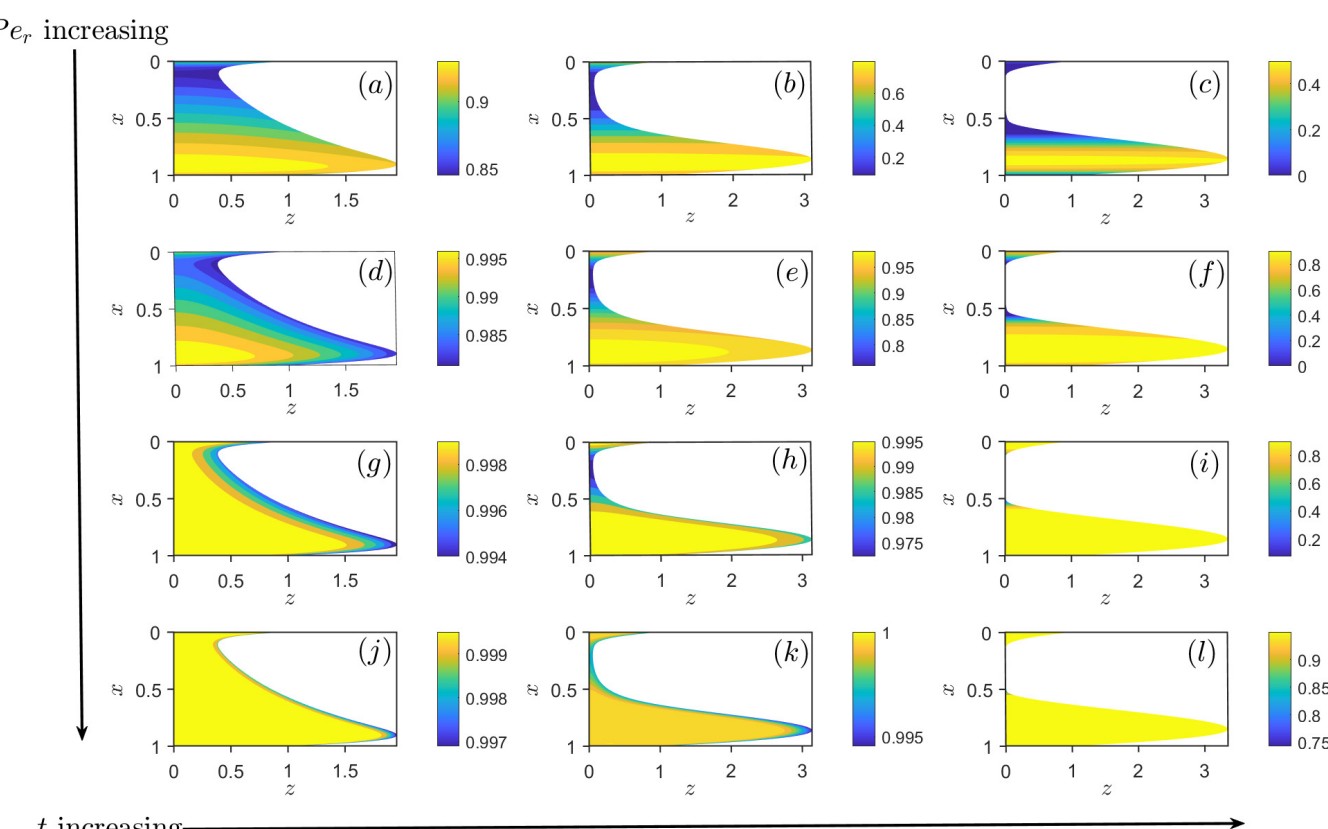

**Figure 3.** The contour plot for (**a**) $\theta(x,z,t=5)$, (**b**) $\theta(x,z,t=20)$ and (**c**) $\theta(x,z,t=100)$ for $Pe_r = 1$; (**d**) $\theta(x,z,t=5)$, (**e**) $\theta(x,z,t=20)$ and (**f**) $\theta(x,z,t=100)$ for $Pe_r = 10$; (**g**) $\theta(x,z,t=5)$, (**h**) $\theta(x,z,t=20)$ and (**i**) $\theta(x,z,t=100)$ for $Pe_r = 10^2$; (**j**) $\theta(x,z,t=5)$, (**k**) $\theta(x,z,t=20)$ and (**l**) $\theta(x,z,t=100)$ for $Pe_r = 10^3$. The other parameter values kept fixed are: $\alpha = 2$, $a = 0.02$, $Ca = 10^3$ and $Re = 0$.

For small $Pe_r$ (dashed curves in Figure 2a), the cooling is significant over the entire film resulting in the temperature quickly dropping to its equilibrium value $\theta = 0$ (or $T^* = T_a^*$), and the evolution of $h(x,t)$ is similar to that of isothermal draining with $\mu(\theta) = 1$ (dashed curves in Figure 2a). For intermediate $Pe_r$ (Figure 2c with $Pe_r = 10, 10^2$, respectively), the cooling is less uniform and pronounced in the thinner lamella section of the film while the temperature is much higher in the thicker pendant drop and upper meniscus regions; the overall viscosity of the liquid is lower than that for low $Pe_r$ leading to faster extensional

flow speed as $Pe_r$ increases (Figure 2d) and hence faster draining and thinning of the lamella region. For much larger $Pe_r$ (dashed curves in Figure 2d,e with $Pe_r = 10^3$), the cooling is confined to a skin near the film's free surface (a diffusive boundary layer) and a collar of cooler liquid forms in the lamella region, with the rest of the liquid within the film insulated at a higher temperature $\theta \approx 1$. This results in a much lower overall viscosity, and consequently faster draining and thinning compared to lower values of $Pe_r$. The evolution of $h(x,t)$ is almost indistinguishable from that of isothermal draining with $\mu(\theta) = \mu_{min}$ (solid curves in Figure 2e).

Figure 2g tracks the evolution of the minimum in $h$, $h_{min}$, as a function of $t$ for $Pe_r$ between $10^{-1} \leq Pe_r \leq 10^3$. $h_{min}$ is representative of the thickness of the lamella film region. We observe increased thinning of the minimum film thickness, $h_{min}(t)$, as $Pe_r$ increases. As $Pe_r$ increases, the fluid drains more quickly, which causes the middle section to become thinner sooner and therefore is more likely to rupture at earlier times. We also observe that $h_{min}$ is always bounded by the two isothermal curves corresponding to $\mu(\theta) = 1, \mu_{min}$, respectively, (red dashed curves in Figure 2g) and the thinning rates for small and large $Pe_r$ tend towards these limiting rates ($\propto t^{-2.25}$) [14]. To characterise the time taken for the film to thin, we define a rupture time $t_{rupt}$ as the time taken for the film to drain to a prescribed thickness. In practice, we estimate $t_{rupt}$ to be the time taken until $h_{min}$ reduces to $5 \times 10^{-2}$ of its initial thickness. We observe that the rupture time is almost doubled as $Pe_r \to 0$.

To highlight the temperature variations within the film and the non-uniform cooling as $Pe_r$ is increased, in Figure 3a–l, we show the contour plot for $\theta(x,z,t)$ at times $t = 5$ (Figure 3a,d,g,j), $t = 20$ (Figure 3b,e,h,k) and $t = 100$ (Figure 3c,f,i,l) for $Pe_r = 1, 10, 10^2, 10^3$, respectively. The other parameter values kept fixed are: $\alpha = 2$, $a = 0.02$, $Ca = 10^3$ and $Re = 0$.

For very small $Pe_r$ (not shown here), the heat loss at the free surface results in the temperature dropping from its initial value $\theta = 1$ ($T = T_i$) to its equilibrium value $\theta = 0$ ($T = T_a$) very quickly. At small values of $Pe_r$, the diffusion of temperature across the thickness of the film dominates, i.e., $\theta_{zz}$, resulting in the film cooling uniformly. As $Pe_r$ increases, the diffusion rate is even slower, and is less dominant in suppressing spatial variations in temperature due to non-uniform cooling both along the film (Figure 3a–c for $Pe_r = 1$ and Figure 3d–f for $Pe_r = 10$) as well as within the film (Figure 3d–f for $Pe_r = 10$). This results in more pronounced cooling in the lamella section of the film where $h$ is much smaller compared to near the ends where the temperatures are much higher as $h$ is comparatively larger there. This non-uniformity in the cooling is due to the rate of heat loss being inversely proportional to $h$—the thicker regions of the film retain their heat more compared to the thinner regions, which lose their heat and therefore cool relatively quickly. This non-uniformity in cooling can be clearly observed in Figure 4a,b which shows the evolution of the temperature along the free surface $\theta(x, z = h(x,t), t)$ for $t$ varying between $t = 1$ and $160$ (in steps of 20) and corresponding to $Pe_r = 1, 10$, respectively. For $Pe_r = 1$, we observe the highest temperatures in the pendant drop region followed by the temperatures in the upper meniscus (Figure 4a). For $Pe_r = 10$, the highest temperatures are in the pendant drop and upper meniscus regions, and we start to observe the development of steep temperature gradients between these regions and the lamella region (Figure 4b). Increasing $Pe_r$ further, the spatial variations in $\theta$ are much more pronounced, with cooling in the middle section of the film where $h$ is much smaller compared to near the ends where $h$ is comparatively larger (Figure 3g–i for $Pe_r = 10^2$). At early times, we also observe variations in $\theta$ within the film (Figure 3g), with the film slowly cooling from the free surface. At later times, it appears that $\theta$ is uniform across the film (Figure 3h,i). The large spatial variation in $\theta$ between the ends and the lamella region is clearly observed in Figure 4c, which shows the evolution of the temperature along the free surface $\theta(x, z = h(x,t), t)$ for $t$ varying between $t = 1$ and $160$ (in steps of 20) and corresponding to $Pe_r = 10^2$. For even larger values of $Pe_r$, we clearly observe that the majority of the cooling is in the lamella section of the film, where the film is very thin; the upper capillary meniscus and the pendant drop region at the bottom remain almost insulated at its initial temperature from

the cooler middle section and a thin cooler boundary layer near the free surface (Figure 3j,k for $Pe_r = 10^3$ where the boundary layer is clearly visible; in Figure 3l, the boundary layer is very thin and not resolved here). This is also clearly identified in Figure 4d which shows the evolution of the temperature along the free surface $\theta(x, z = h(x, t), t)$ for $t$ varying between $t = 1$ and 160 (in steps of 20) and corresponding to $Pe_r = 10^3$. The significant reduction in the cooling of the middle lamella section is clearly evident at higher $Pe_r$. This is due to the enhanced convection of heat through the flow coming from the hotter upper meniscus region.

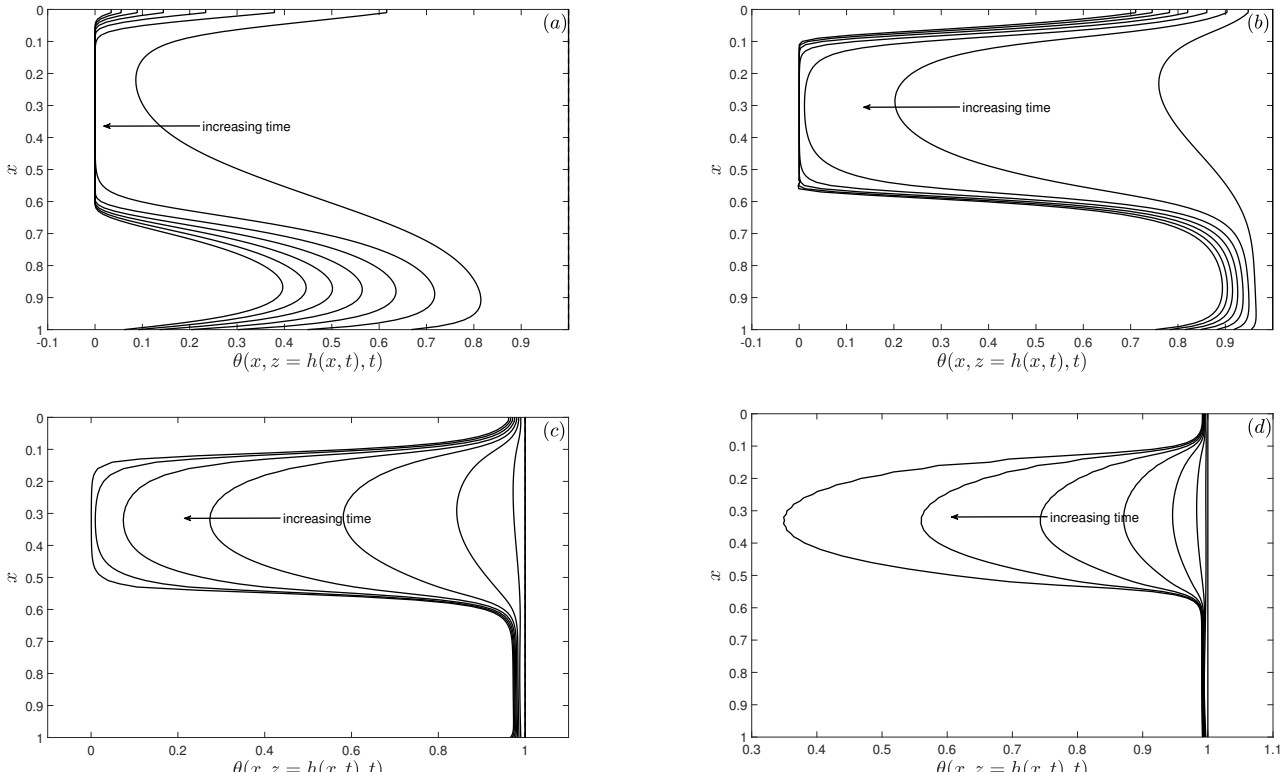

**Figure 4.** The evolution of the temperature at the free surface, $\theta(x, z = h(x, t), t)$ for $t$ varying between $t = 0$–160 (in steps of 20) corresponding to (**a**) $Pe_r = 1$, (**b**) $Pe_r = 10$, (**c**) $Pe_r = 10^2$ and (**d**) $Pe_r = 10^3$. The other parameter values kept fixed are: $\alpha = 2$, $a = 0.02$, $Ca = 10^3$ and $Re = 0$.

Next, we investigate the influence of the viscosity–temperature decay constant $\alpha$, the heat transfer coefficient at the free surface $a$ and the surface tension–temperature parameter $M$ on the global minimum film thickness $h_{min}$.

Figure 5a investigates the influence of varying $\alpha$ on $h_{min}(t)$ for fixed $Pe_r = 10^3$ and $a = 0.02$. We observe the increased thinning of the minimum film thickness $h_{min}(t)$ as $\alpha$ increases. As $\alpha$ increases, the fluid drains more rapidly (due to the larger reduction in viscosity), which accelerates the the thinning of the middle section, therefore lowering the rupture times (by almost half the time compared to the isothermal $\mu = 1$ case). In the limit $\alpha \to 0, \infty$, we recover the isothermal cases corresponding to $\mu = 1, \mu_{min}$, respectively (red dashed curves in Figure 4a). Figure 5b investigates the influence of varying $a$ on $h_{min}(t)$ for fixed $Pe_r = 10^3$ and $\alpha = 2$. We observe that the thinning of the minimum film thickness $h_{min}(t)$ decreases as $a$ increases. The fluid drains more slowly, which slows down the thinning of the lamella section, therefore delaying the rupture times. We now study the influence of varying $M$ on $h_{min}(t)$ for two cases corresponding to a low value of $Pe_r = 1$ (Figure 5c) and a high value of $Pe_r = 10^3$ (Figure 5d). We fix $\alpha = 2$ and $a = 0.02$. For low values of $Pe_r$, we observe $h_{min}$ to marginally increase with $M$; the increase is exaggerated for larger values of $M$ (Figure 5c). This is due to gradients in surface tension generated due to variations in $\theta$ along the film (i.e., $\theta_x$), which is much stronger in the transition region

between the downstream end of the lamella region and the pendant drop compared to the transition region between its upstream end and the upper meniscus region (see Figure 4a). Moreover, the stronger surface tension gradients at the downstream end of the lamella region oppose the gravity-driven flow, hence slowing down the extensional flow speed and thereby reducing the thinning of the lamella region.

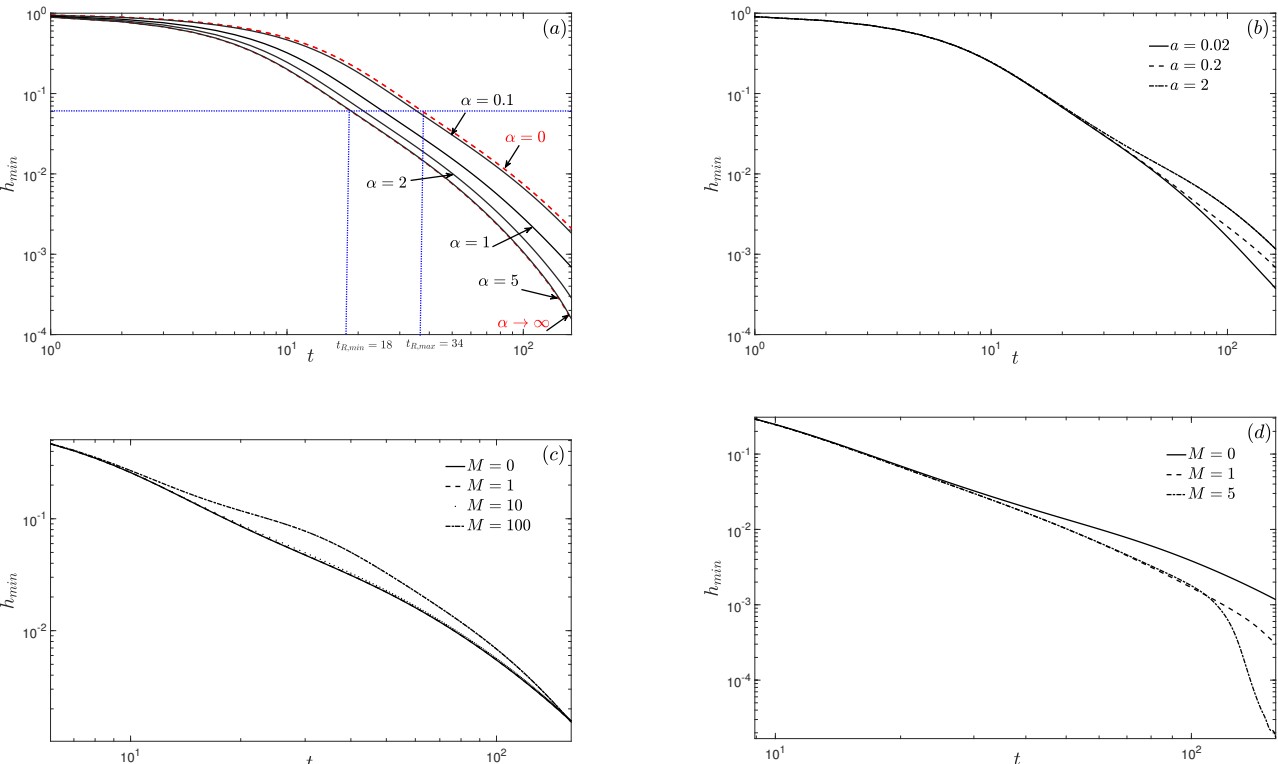

**Figure 5.** The global minimum $h_{min}$ as a function of time $t$ for (**a**) varying $\alpha$ ($Pe_r = 10^3$, $a = 0.02$), (**b**) varying $a$ ($Pe_r = 10^3$, $\alpha = 2$), (**c**) varying $M$ ($Pe_r = 1$, $a = 0.02$) and (**d**) varying $M$ ($Pe_r = 10^3$, $a = 0.02$). The other parameter values kept fixed are $Ca = 10^3$ and $Re = 0$.

In contrast, for high values of $Pe_r$, we observe a decrease in $h_{min}$ at late times as $M$ increases; the drop in $h_{min}$ is quite dramatic for higher values of $M$. In this case, the surface tension gradients in the transition region between the upstream end of the lamella and the upper meniscus region are stronger than that in the transition region between its downstream end and the pendant drop region (due to $\theta_x$ being larger at the upstream end; see Figure 4d). This contribution cooperates with the gravity-driven flow, hence increasing the extensional flow speed and thereby accelerating the thinning of the lamella region.

## 5. Discussion

In this paper, we coupled the thin-film flow equations to a two-dimensional advection-diffusion equation for the temperature field and investigated the draining and thinning of a cooling liquid film. We considered non-isothermal conditions which included a temperature-dependent viscosity and surface tension and heat loss due to cooling at the free surface. A systematic parameter study revealed the influence of the system parameters on this cooling, particularly the reduced Péclet number $Pe_r$, the decay constant in the exponential viscosity–temperature model $\alpha$, the heat transfer coefficient $a$, and the slope of the linear surface tension–temperature model $M$. The resulting temperature and corresponding viscosity and surface tension contrast arising due to the cooling near the film's free surface significantly influenced the draining and subsequent thinning of the film.

A key contribution of this work distinguished the thinning rate and rupture times of the lamella between the non-isothermal cases studied here and the isothermal cases from our previous work [14]. Indeed, we demonstrated the significant influence of cooling and showed that the lamella can thin and rupture either faster or slower than the corresponding isothermal cases (Figures 2g and 5). This was dependent on the parameter values.

The main highlight of our results identified an important feature during the draining and thinning process—the preferential cooling in the film's flat middle section (lamella) compared to the top and bottom regions (Plateau borders). The rate of heat loss in the lamella was maximum due to its much smaller thickness compared to the much thicker Plateau borders (Figure 4). The extent of this cooling was dependent on the parameter values, in particular the reduced Péclet number $Pe_r$. For intermediate and large $Pe_r$, a draining *collar* of colder liquid was observed in the lamella sandwiched between two much hotter Plateau border regions. The hotter regions appeared to be almost insulated from the cooler middle section and a thin cooler boundary layer near the free surface (Figures 3i,l and 4c,d). In contrast, for small values of $Pe_r$, the temperature isotherms are almost constant across the film thickness (Figure 3a–c) and the film cooled almost uniformly along its thickness. The non-uniform cooling and its influence on foam film drainage identified in our work clearly suggests that it is necessary to include the heat transfer and drainage both in the lamella and Plateau borders, which was not considered in previous work [7]. Moreover, the cooling of the free surface was also shown to be important, which was neglected in previous work [7]. In our model, we have assumed that the wire frames are insulated; future work will include heat transfer from both the free surface and wire frames.

We observed that the cooling rate could be enhanced by increasing the heat transfer coefficient $a$ which slowed down the draining and thinning of the film. Moreover, a rapid drop in the viscosity with temperature controlled by the parameter $\alpha$ increased the draining flow and the subsequent thinning of the film. The low $Pe_r$ limit is preferred in metallic films since the hot liquid in the film cools uniformly and rapidly. Consequently, the liquid viscosity increases uniformly within the film, resulting in slower drainage and thinning of the film. This can be achieved if the Péclet number $Pe = U^*L^*/\kappa_d^*$ is small (equivalently if the thermal diffusivity for the liquid $\kappa_d^*$ is large or the aspect ratio $\epsilon$ is small). For melts with low diffusivity, one would need very thin films for the low $Pe_r$ results to be achieved. Another method to sufficiently reduce the drainage so that cooling can occur is to disperse particles within the melt that can increase its effective viscosity, e.g., alumina particles are dispersed in aluminium foam to increase the viscosity [5,6].

Our investigations on the influence of temperature variations in surface tension showed that effect of increasing the slope of the linear surface tension–temperature relationship $M^*$ was observed to be more effective at lower Péclet numbers. In this parameter range, surface tension gradients in the lamella region opposed the gravity-driven flow. At higher Péclet numbers, though, the surface tension gradients tended to enhance the draining flow in the lamella region resulting in the dramatic thinning of the film at late times. Our results indicated that the thermocapillary effect had much less influence on the draining and thinning of the film in comparison to thermoviscous effects. This was due to a limitation in our model which restricted the variation in surface tension with temperature to $O(\epsilon^2)$ in order to relegate the influence of surface tension gradients to $O(\epsilon^2)$. To accommodate larger variations in surface tension, this needs to be relaxed, and a different dominant balance, including surface tension gradients at the leading order in $\epsilon$, will need to be explored.

A major limitation of this study was not considering the influence of phase transition due to solidification. This limited our results to be only valid for temperatures much larger than the melting temperature. We were unable to investigate scenarios where, for example, a solid crust forms at the air–liquid interface and its interaction with the hot liquid core [7]. As part of future work, we will need to modify the viscosity–temperature relationship in Equation (16a) to model the change in viscosity at temperatures close to the melting point,

e.g., Cox et al. [7] chose a step function for $\mu$ that gives small values at high temperatures and high values at low temperatures. In addition, the latent heat of fusion will need to be considered. Cox et al. [7] use a simple specific heat–temperature relationship to mimic a peak in the specific heat around the melting temperature to represent the heat that must be absorbed before the foam solidifies. Incorporating these relationships into our model will allow us to fully describe the cooling and solidification of metallic foam films.

The theoretical framework developed here is versatile and can be readily adapted to accommodate complex melts exhibiting non-Newtonian or viscoelastic behaviour with temperature-dependent properties. This insight would form the basis for future developments of this model to investigate the overall behaviour of a foam network, for example, using the framework proposed by Stewart et al. [24].

**Author Contributions:** Conceptualization, H.A. and S.N.; methodology, H.A. and S.N.; formal analysis, H.A.; investigation, H.A.; writing—original draft preparation, S.N.; writing—review and editing, S.N.; supervision, S.N. All authors have read and agreed to the published version of the manuscript.

**Funding:** This research received no external funding.

**Data Availability Statement:** The data that support the findings of this study are available from the corresponding author upon reasonable request.

**Acknowledgments:** This work was a part of Hani Alahmadi's PhD research at Keele University. Hani gratefully acknowledges financial support from Jouf University, the Ministry of Education, Kingdom of Saudi Arabia, and the Saudi Arabian Cultural Bureau in London (UKSACB).

**Conflicts of Interest:** The authors declare no conflict of interest.

## Appendix A. Derivation of the PDEs in (20)

We exploit the fact that $\epsilon = \dfrac{H_0^*}{L^*} \ll 1$ and expand each of the unknowns variables $(u, w, p, \tau^{xx}, \tau^{zz}, \tau^{xz}, h)$ as a power series in $\epsilon^2$ of the form:

$$(u, w, p, \tau^{xx}, \tau^{zz}, \tau^{xz}, \theta) = (u, w, p, \tau^{xx}, \tau^{zz}, \tau^{xz}, \theta)_0(x, z, t) + \epsilon^2 (u, w, p, \tau^{xx}, \tau^{zz}, \tau^{xz}, \theta)_1(x, z, t) + O(\epsilon^4). \tag{A1}$$

Substituting this into Equations (18a)–(18n). we obtain at $O(1)$:

$$u_{0x} + w_{0z} = 0, \tag{A2}$$

$$\tau_{0z}^{xz} = 0, \tag{A3}$$

$$-p_{0z} + \tau_{0x}^{xz} + \tau_{0z}^{zz} = 0, \tag{A4}$$

$$w_0 = u_{0z} = \tau_0^{xz} = 0, \text{ at } z = 0, \tag{A5}$$

$$-p_0 + \tau_0^{zz} - 2h_x \tau_0^{xz} = \frac{1}{Ca} h_{xx}, \text{ at } z = h \tag{A6}$$

$$\tau_0^{xz} = 0, \text{ at } z = h. \tag{A7}$$

Equations (A3), (A5) and (A7) imply that

$$\tau_0^{xz}(x, z, t) = 0. \tag{A8}$$

Integrating Equation (A4) with respect to $z$ and using Equation (A5) and (A6), we obtain

$$p_0 = \tau_0^{zz} - \frac{1}{Ca} h_{xx}. \tag{A9}$$

To determine $\tau_0^{xx,zz}$, we need to analyse the $O(\epsilon^2)$ equations. Before we do this, we note the following: $u_{0z} = 0$, so $u_0 = u_0(x, t)$, using $\tau_0^{xz} = 0$ and Equation (18e) at leading order. In addition, $\tau_0^{zz} = -\tau_0^{xx}$, using Equation (A2) in Equation (18e). Equation (A2) also

gives $w_{0z} = -u_{0x}$, which on integrating with respect to $z$ and using $w_0 = 0$ at $z = 0$, gives $w_0(x, z, t) = -u_{0x}z$. At $O(\epsilon^2)$, we have

$$Re(u_{0t} + u_0 u_{0x} + w_0 u_{0z}) = -p_{0x} + \tau_{0x}^{xx} + \tau_{1z}^{xz} + 1, \tag{A10}$$

$$Re(w_{0t} + u_0 w_{0x} + w_0 w_{0z}) = -p_{1z} + \tau_{1x}^{xz} + \tau_{1z}^{zz}, \tag{A11}$$

$$w_1 = u_{1z} = \tau_{1z}^{xz} = 0, \text{ at } z = 0, \tag{A12}$$

$$\tau_1^{xz} - h_x^2 \tau_0^{xz} + h_x(\tau_0^{zz} - \tau_0^{xx}) = -\frac{M}{Ca}[\theta_{0x} + h_x \theta_{0z}], \text{ at } z = h. \tag{A13}$$

Integrating Equation (A10) with respect to $z$ and using Equation (A12), we obtain

$$\tau_1^{xz} = -2\int_0^z \tau_{0x}^{xx} dz - \left[\frac{1}{Ca}h_{xxx} + 1 - Re(u_{0t} + u_0 u_{0x})\right] z. \tag{A14}$$

Substituting this into Equation (A13) gives

$$2\int_0^h \tau_{0x}^{xx} dz + 2h_x \tau_0^{xx} + h\left[\frac{1}{Ca}h_{xxx} + 1 - Re(u_{0t} + u_0 u_{0x})\right] = \frac{M}{Ca}[\theta_{0x} + h_x \theta_{0z}|_{z=h}]. \tag{A15}$$

Equation (A15) represents the force balance at the free surface of the extensional stress (represented by the first two term), surface tension (represented by the third term), gravity (represented by the fourth term), inertia (represented by the fifth and sixth terms) and variations in surface tension (represented by the last term).

To determine the evolution equation of $h$ using Equation (18j), we also need to determine $u_0$ and the $O(\epsilon^2)$ correction $u_1$. We use the constitutive law to determine these. From Equation (18e), we obtain

$$u_{0x} = \frac{1}{2\mu(\theta_0)}\tau_0^{xx}, \tag{A16}$$

$$u_{1z} + w_{0x} = \frac{1}{\mu(\theta_0)}\tau_1^{xz} \Rightarrow u_{1z} = \frac{1}{\mu(\theta_0)}\tau_1^{xz} - w_{0x} = \frac{1}{\mu(\theta_0)}\tau_1^{xz} + u_{0xx}z, \tag{A17}$$

where $\mu(\theta_0)$ is given by Equation (18n). We can combine Equations (A15) and (A16) to write a single evolution equation for $u_0$. This can be written as:

$$Re\, h(u_{0t} + u_0 u_{0x}) - 4\left[\mu(\theta_0)h_x u_{0x} + \int_0^h (\mu(\theta_0)u_{0x})_x\, dz\right] = h\left[\frac{1}{Ca}h_{xxx} + 1\right] - \frac{M}{Ca}[\theta_{0x} + h_x \theta_{0z}|_{z=h}]. \tag{A18}$$

Finally, the evolution equation for $h$ can be obtained from Equation (18j) as:

$$h_t + Q_{0x} = 0, \quad Q_0 = u_0 h. \tag{A19}$$

Hence, Equations (A18) and (A19) provide a coupled system of two PDEs for the film's free surface evolution, $h(x, t)$ and the extensional flow speed $u_0(x, t)$, respectively.

**Appendix B. Mapping $(x, z) \in [0, 1] \times [0, h]$ to a Rectangular Domain $(x, z) \in [0, 1] \times [0, 1]$**

In order to solve Equation (20) numerically, it is instructive to map $(x, z) \in [0, 1] \times [0, h]$ to a rectangular domain $(x, z) \in [0, 1] \times [0, 1]$. We apply the following change of variables:

$$\bar{x} = x, \quad \bar{z} = \frac{z}{h(x, t)}, \quad \bar{t} = t. \tag{A20}$$

Using the chain rule, we can write

$$\frac{\partial}{\partial x} = \frac{\partial}{\partial \bar{x}} - \frac{\bar{z}h_{\bar{x}}}{h}\frac{\partial}{\partial \bar{z}}, \quad \frac{\partial}{\partial z} = \frac{1}{h}\frac{\partial}{\partial \bar{z}}, \quad \frac{\partial}{\partial t} = \frac{\partial}{\partial \bar{t}} - \frac{\bar{z}h_{\bar{t}}}{h}\frac{\partial}{\partial \bar{z}}. \tag{A21}$$

Applying the above change of variables to Equations (20d)–(20f), we obtain the transformed evolution equation for $\theta(\bar{x}, \bar{z}, \bar{t})$ given by

$$Pe_r\left[\theta_{\bar{t}} + u\theta_{\bar{x}} + (w - \bar{z}uh_{\bar{x}} - \bar{z}h_{\bar{t}})\frac{1}{h}\theta_{\bar{z}}\right] = \frac{1}{h^2}\theta_{\bar{z}\bar{z}} + \epsilon^2\left[\theta_{\bar{x}\bar{x}} - \bar{z}\left(\frac{h_{\bar{x}}}{h}\right)_{\bar{x}}\theta_{\bar{z}} - \frac{\bar{z}h_{\bar{x}}}{h}\left(2\theta_{\bar{x}\bar{z}} - \left(\frac{\bar{z}h_{\bar{x}}}{h}\theta_{\bar{z}}\right)_{\bar{z}}\right)\right], \ (\bar{x}, \bar{z}) \in [0,1] \times [0,1],$$

$$w(\bar{x}, \bar{z}, \bar{t}) = -u_{\bar{x}}h\bar{z}, \ (\bar{x}, \bar{z}) \in [0,1] \times [0,1], \tag{A22a}$$

$$\theta_{\bar{z}} = 0, \text{ at } \bar{z} = 0, \ \forall \bar{x} \in [0,1], \ \ \theta_{\bar{z}} = -a\epsilon^2 h\theta, \text{ at } \bar{z} = 1, \ \forall \bar{x} \in [0,1], \tag{A22b}$$

$$\theta_{\bar{x}} = \epsilon^2 b(\theta - \theta_s) + \frac{\bar{z}h_{\bar{x}}}{h}\theta_{\bar{z}}, \text{ at } \bar{x} = 0, \ \forall \bar{z} \in [0,1], \ \ \theta_{\bar{x}} = -\epsilon^2 b(\theta - \theta_s) + \frac{\bar{z}h_{\bar{x}}}{h}\theta_{\bar{z}}, \text{ at } \bar{x} = 1, \ \forall \bar{z} \in [0,1]. \tag{A22c}$$

The film thickness evolution, Equation (20a), and the extensional flow speed evolution, Equation (20b), in the transformed coordinates become,

$$h_{\bar{t}} + Q_{\bar{x}} = 0, \ Q = uh, \tag{A23a}$$

$$Reh(u_{\bar{t}} + uu_{\bar{x}}) - 4\left[\mu(\theta)h_{\bar{x}}u_{\bar{x}} + \int_0^1(\mu(\theta)u_{\bar{x}})_{\bar{x}}h\,d\bar{z} - \int_0^1 \bar{z}h_{\bar{x}}(\mu(\theta)u_{\bar{x}})_{\bar{z}}\,d\bar{z}\right] = h\left[\frac{1}{Ca}h_{\bar{x}\bar{x}\bar{x}} + 1\right] - \frac{M}{Ca}\left[\theta_{\bar{x}} + \frac{h_{\bar{x}}}{h}\theta_{\bar{z}}|_{\bar{z}=1}\right]. \tag{A23b}$$

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
