# Peer review of "The Role of Thermoviscous and Thermocapillary Effects in the Cooling and Gravity-Driven Draining of Molten Free Liquid Films"

_fluids, doi:10.3390/fluids8050153_

Round 1

Reviewer 1 Report

The paper studies the drainage of a thin free fluid film between rigid frames with heat transfer cooling the warm film in a cool passive environment.  The film models metals by having high conductivity, leading to small Peclet number and an appropriate viscosity depending on temperature.  The fluid dynamics is governed by a thin film approximation, but the temperature field remains 2D. The thin film problem is solved in a 1D fixed grid, and the thermal problem is solved on a fixed rectangular domain to which that problem is mapped.  The solutions appear to be correct and the results presented well. I enjoyed reading the paper.

I have some points to make for minor revisions.

1.      1. The Results section’s paragraphs 2 and 3 are about the numerical method.  I recommend moving those out to a Numerical Methods section as the last subsection prior to the Results section.  I also request that the mapped equations be reported in an appendix as the mapping is non-trivial here.  It would be helpful to completely state the numerical system that is actually solved.

2.     2.  In the expansions given in equation (19), and (A1), is the correction to h needed?

3.      3. In Figure 3 and similar figures, the lower right corner has a scale that is invisible in the contour plot itself. Could an inset show the variation in the thin region? If the solution is not valid in that area, can an earlier time with an inset be used?

4.      4. In the discussion section, second paragraph, the first sentence and item (i) are very long.  Please break them up and make them simple to read; these are important points and it will help the reader understand them.

5.     5.  Line 388: is a closing parenthesis needed between “large” and “or”?

6.     6.  In (A14) and (A18) please make parentheses and brackets the correct larger size for the contents between them. This goes for all equations in the paper.

Author Response

We thank the referee for their very useful and helpful comments. We have taken all the comments on board. We hope that the referee is satisfied with the revisions. All changes to the manuscript are highlighted in bold.

We now address each point made by the referee.

  1. We agree with the referee and have now included a new section - Section 3. Numerical Methods. We have now included appendix B which reports the mapped equations. We have decided against stating the complete numerical system as it is quite long and shifts focus from the physical problem itself. We have included a new reference which is the PhD thesis of one of the authors where the numerical system is stated in detail.
  2. We have removed the connection to h in the expansion given in Eqs. (19,A1).
  3. We do agree with the referee but have decided against showing an inset because it would make the figure crowded. In fact, Figs. 4(c,d) show the variation of the temperature along the film's free surface which clearly shows the variation in the thin middle section in comparison to the top and bottom. The corresponding text also describes this in some detail. 
  4. We have now modified the Discussion section to reflect the referee's comment as well as highlight the novel contributions and limitations of the study requested by another referee.
  5. Done.
  6. Done.

Thank you.

Reviewer 2 Report

    The research work conducted in this study is a good contribution to the literature on the selected topic. However, the following revisions are required before the publication of this manuscript:

    1. What are the limitations of the defined problem?

    2. Relations 16 (a&b) and dimensionless variables (17) should be referenced. 

    3. How the authors selected the physical parameters in this study? What are the ranges of these parameters? 

    4. The results of the present study should be compared with the results of earlier works and/or other methodologies, experimental results or numerical results as limiting cases.

    5. The conclusions are not as expected. This section should be reduced to the novel contributions due to the method/problem investigated.

    6. Add units/dimensions of all physical parameters in the nomenclature. 

Author Response

We thank the referee for their helpful comments. We have taken on board all their comments. We hope that the referee is satisfied with the revisions. All the revisions are highlighted in bold in the revised manuscript.

We now address each point raised by the referee.

  1. We have now included a key limitation in the Discussion section.
  2. We have now provided references for Eq. (16a,b) and dimensionless variables in Eq. (17).
  3. We have now provided 2 tables (Table 1 and 2) with the physical quantities and the dimensionless parameters appearing in our model. Some of the liquid melt properties are based on Aluminium foam melts, while some other values have been assumed as they are difficult to obtain from literature,
  4. We have now made comparisons with our previous work (Ref 1) and a previous work investigating non-isothermal drainage in foams (Ref 8) in the Discussion section.
  5. We have revised the Discussions section to highlight our important contributions.
  6. This has been done as described in point 3 above.

Thank you.